A new species of frog of the genus Noblella Barbour, 1930 (Amphibia: Strabomantidae) from the Cordillera del Cóndor, Ecuador

Brito-Zapata David fredavidbrito@gmail.com 1
Chávez-Reyes Juan D. 1
Pallo-Robles Matheo David 1
Carrión-Olmedo Julio C. 2
Cisneros-Heredia Diego F. 1 2
Reyes-Puig Carolina 1 2
1 Museo de Zoología, Laboratorio de Zoología Terrestre, Instituto de Biodiversidad Tropical IBIOTROP, Colegio de Ciencias Biológicas y Ambientales, Universidad San Francisco de Quito , Quito , Pichincha , Ecuador
2 Unidad de Investigación, Instituto Nacional de Biodiversidad , Quito , Pichincha , Ecuador
Sunny Armando
Electronic publication date: 2024 Oct 2
Publication date: 2024
Volume: 12
Electronic Location ID: e17939
Received 2024 Apr 9; Accepted 2024 Jul 26
Copyright: ©2024 Brito-Zapata et al.
Copyright year: 2024
Copyright holder: Brito-Zapata et al.
License: This is an open access article distributed under the terms of the Creative Commons Attribution License, which permits unrestricted use, distribution, reproduction and adaptation in any medium and for any purpose provided that it is properly attributed. For attribution, the original author(s), title, publication source (PeerJ) and either DOI or URL of the article must be cited.
License URL: https://creativecommons.org/licenses/by/4.0/

Keywords: Leaflitter frog, Osteology, Phylogeny, Taxonomy, Zamora Chinchipe

Funding: Universidad San Francisco de Quito USFQ Instituto Nacional de Biodiversidad This research was supported by Universidad San Francisco de Quito USFQ through operative funds assigned to the Instituto de Biodiversidad Tropical IBIOTROP and by the Instituto Nacional de Biodiversidad through funds provided by Staedtler Forestal Staforco Cia. Ltda. and GAD El Oro. The funders had no role in study design, data collection and analysis, decision to publish, or preparation of the manuscript.

==============================
We describe a new species of leaflitter frog of the genus Noblella from southern Ecuador, province of Zamora Chinchipe. The new species is diagnosed from all its congeners by having one or two tubercles on the upper eyelids; distal phalanges strongly T-shaped; phalangeal formula on hands 2-2-3-3; phalangeal formula on feet 2-2-3-4-3; heel with a small subconical tubercle; disc on all toes with papillae; dorsum brown or brown with gray, with V-shaped inverted or scattered irregular darker marks. We include a detailed description of its osteology and a study of its phylogenetic relationships. Finally, we evaluate its conservation status and discuss the threats that are currently impacting at the type locality.

Introduction

The tropical Andes exhibit a high species diversity, with amphibians standing out as one of the richest taxonomic groups in the region (Duellman, 1999; Myers et al., 2000; Hutter, Lambert & Wiens, 2017; Luedtke et al., 2023). Over the past two decades in the tropical Andes, the number of species of amphibians has significantly increased thanks to the taxonomic work conducted especially in Colombia, Ecuador, and Peru (e.g., Castroviejo-Fisher et al., 2009; Reyes-Puig et al., 2010; Reyes-Puig et al., 2014; Reyes-Puig et al., 2019; Brito & Pozo-Zamora, 2013; Brito, Batallas & Yánez-Muñoz, 2017; Páez & Ron, 2019; Cisneros-Heredia et al., 2023). Ecuador plays a pivotal role in this biodiversity hotspot, being one of the three countries with the highest amphibian diversity in the world (Ortega-Andrade et al., 2021; Frost, 2024; AmphibiaWeb, 2024).

The topographical intricacies of Ecuador, with multiple mountain ranges of different geological ages and hydrographical features with Pacific and Amazonian drainages, have promoted an evolutionary explosion of biodiversity in the country (Richter et al., 2009; Novotná et al., 2018; Vélez-Abarca et al., 2023). The Cordillera del Cóndor is a subAndean mountain range located at the geopolitical boundary between southeastern Ecuador and northern Peru. It is characterized by rugged terrains reaching a maximum elevation of about 2900 m and primarily constituted by Mesozoic and early Cenozoic sandstones and limestones deposited before the emergence of the Andes (Neill, 2005). The intricate interplay of geological and ecological factors in the Cordillera del Cóndor has fostered the biological richness of the region, being a hotspot of diversity and endemicity (Neill, 2005; Novotná et al., 2018; Vélez-Abarca et al., 2023).

The first modern herpetological expeditions to the Cordillera del Cóndor started in the 1970s and immediately showed a vast number of range-restricted amphibian species (Lynch, 1974a; Lynch, 1974b; Lynch, 1976; Lynch & Duellman, 1980; Duellman & Lynch, 1988; Duellman & Simmons, 1988). Over the last few years, studies on the amphibians of the Cordillera del Cóndor have increased, and many new endemic species have been described (Cisneros-Heredia & Morales, 2008; Jungfer, 2011; Almendariz, Ron & Brito, 2012; Almendariz, Brito & Batallas, 2014; Valencia et al., 2017; Brito-Zapata & Reyes-Puig, 2021; Brito-Zapata et al., 2021; Székely et al., 2023; Cisneros-Heredia et al., 2023). Paradoxically, despite its ecological significance, this region remains continuously threatened by anthropogenic activities, mainly habitat loss and fragmentation due to mining and expansion of the agricultural frontier (Chicaiza & Yánez, 2012; Fernández-Salvador, Teijlingen & Leifsen, 2017; Roy et al., 2018; Brito-Zapata et al., 2021).

The genus Noblella Barbour 1930 is a group of minute frogs of cryptic, terrestrial habits. There are 17 formally described species of Noblella, most of them with small geographic ranges, distributed in the Andes of Ecuador, Peru, and Bolivia and the Amazonia of southeastern Colombia, Ecuador, Peru, Bolivia, and Brazil (Reyes-Puig et al., 2021; Frost, 2024). Previous studies have revealed the polyphyly of Noblella, with one clade (termed as Southern Clade) from the Andes of central and southern Peru and Bolivia most closely related to the genera Psychrophrynella, Microkayla and Qosqophryne and another clade from the northern Andes and Amazonia, where the Ecuadorian species are included (termed as the Northern Clade) (Catenazzi & Ttito, 2019; Reyes-Puig et al., 2019; Reyes-Puig et al., 2020; Reyes-Puig et al., 2021; Santa-Cruz et al., 2019). Although taxonomic resolutions to this issue are still pending, the diversity of Noblella sensu lato keeps increasing with the discovery of undescribed taxa. Herein, we describe a new species of the northern clade of Noblella from the Cordillera del Cóndor, Ecuador, based on morphological, osteological, and molecular evidence.

Materials & Methods

Ethical statement

This study was conducted under research permits (MAAE-ARSFC-2022-2204, 009-2018-IC -FLO-FAU-DPAZCH-UPN-VS/MA, 005-2019-IC-FLO-FAU-DPAZCH-UPN-VS/MA) and framework contract for genetic resources access (MAATE-DBI-CM-2023-0313) issued by the Ministerio del Ambiente, Agua y Transición Ecológica, Ecuador. Ethical use of live amphibians and reptiles in field research followed the guidelines outlined by Beaupre et al. (2004).

Species concept

We adhere to the unified species concept proposed by De Queiroz (2007), defining species as independently evolving metapopulation lineages discernible from an operational standpoint by inferring isolation from their putative sister lineages.

Taxonomic sampling

Specimens from the following collections were examined: Instituto Nacional de Biodiversidad, Quito (DHMECN); National Museum of Natural History, Smithsonian Institution, Washington, D.C. (USNM); Museo de Zoología, Universidad San Francisco de Quito, Quito (ZSFQ). Information for comparison between species was obtained from the literature (Noble, 1921; Lynch, 1976; Lynch, 1986; Duellman, 1991; De la Riva & Köhler, 1998; Köhler, 2000; Lehr, Aguilar & Lundberg, 2004; Duellman & Lehr, 2009; Lehr & Catenazzi, 2009; Guayasamin & Terán-Valdez, 2009; Harvey et al., 2013; Catenazzi, Uscapi & May, 2015; Catenazzi & Ttito, 2019; Reyes-Puig et al., 2019; Reyes-Puig et al., 2020; Reyes-Puig et al., 2021; Santa-Cruz et al., 2019) and the following specimens (H = holotype): Noblella coloma (Guayasamin & Terán-Valdez, 2009): ECUADOR: Santo Domingo de los Tsáchilas: ZSFQ 3266; Noblella mindo (Reyes-Puig et al., 2021): ECUADOR: Pichincha: ZSFQ 050 (H), 049, 051, 304-305; Noblella lochites (Lynch, 1976): ECUADOR: Napo: ZSFQ 346–348, Zamora Chinchipe: ZSFQ 1119, 1124, 1186–1188; Noblella myrmecoides (Lynch, 1976): PERU: Loreto: USNM 127181, 332985; Noblella naturetrekii (Reyes-Puig et al., 2019): ECUADOR: Tungurahua: DHMECN 13390 (H), 13307, 13391, 14411, 14420, 14493–4, ZSFQ 934; Noblella worleyae (Reyes-Puig et al., 2020): ECUADOR: Imbabura: ZSFQ 345, 550–2, 2502–4, 3851–2. Several studies have shown phylogenies with unrelated clades reported as Noblella myrmecoides, and distributed in the Amazonian lowlands in Ecuador, Bolivia, Peru, and Brazil (Duellman & Lehr, 2009; Harvey et al., 2013; Reyes-Puig et al., 2019; Reyes-Puig et al., 2020; Reyes-Puig et al., 2021). Noblella lochites have been reported from many locations in the eastern Andes and Subandean cordilleras of Ecuador and Peru (Lynch, 1976; Duellman & Lynch, 1988; Almendariz et al., 2014). Therefore, to diagnose our new species, we restricted our comparisons to the definitions provided in their original type descriptions.

The electronic version of this article in Portable Document Format (PDF) will represent a published work according to the International Commission on Zoological Nomenclature (ICZN), and hence the new names contained in the electronic version are effectively published under that Code from the electronic edition alone. This published work and the nomenclatural acts it contains have been registered in ZooBank, the online registration system for the ICZN. The ZooBank LSIDs (Life Science Identifiers) can be resolved and the associated information viewed through any standard web browser by appending the LSID to the prefix http://zoobank.org/. The LSID for this publication is: LSID urn:lsid:zoobank.org:pub:AD15D34E-81A6-4666-AF30-E53D4F3C5541. The online version of this work is archived and available from the following digital repositories: PeerJ, PubMed Central SCIE and CLOCKSS.

Fieldwork

Fieldwork was conducted at the Río Blanco community, close to the Paquisha military detachment, western slope of the Cordillera del Cóndor (3.8942°S, 78.5166°W, 1560–2403 m elevation), Paquisha parish, Zamora Chinchipe province, Ecuador (Fig. 1). Field expeditions were carried out in 2018, 2019, and 2023. We surveyed amphibians and reptiles during the day (09h00–11:00) and night (19h00–23:00), conducting detailed searches in the forest ground, ground holes, leaf litter, and decomposing trunks. We used five 20 L pitfall traps separated by 2.5 m along a straight line, an effective technique for collecting Noblella (Reyes-Puig et al., 2019). Specimens were taken to the field station in plastic bags, photographed alive, and euthanized with lidocaine; muscle or liver tissue samples were extracted and preserved in 95% ethanol at −20 °C, and whole specimens were fixed in 8% formalin and preserved in 75% ethanol. Specimens and tissue samples are deposited at the ZSFQ.

Figure 1 Map showing the type locality of Noblella arutam sp. nov. (star). Ecuador in color. Map by Emilia Peñaherrera.

External morphology and coloration

Diagnosis and description were based on formats proposed by Lynch & Duellman (1997), Duellman & Lehr (2009), and Reyes-Puig et al. (2021). All characters reported are from adult specimens. To determine sex and maturity, we directly inspected the gonads through a dorsoventral incision. We measured preserved specimens with a digital caliper (0.05 mm accuracy, rounded to the nearest 0.1 mm). We followed the main diagnostic characters proposed by Reyes-Puig et al. (2021) for comparisons with other species. We took the following measures: snout-vent length (SVL); head length (HL); head width (HW); horizontal diameter of eye (ED); eye-nostril distance (EN); horizontal diameter of tympanum (TD); minimum interorbital distance (MIOD), taken between the inner edges of the eyelids; minimum eyelid width (MWE), taken from the inner border to the outer border of the eyelid; hand length (HaL); shank length (SL); foot length (FL).

Osteology

The osteological description was based on an adult male specimen (ZSFQ 1886). We followed the protocol Tsandev et al. (2017) established for the transparency process with modifications and specifications detailed below. The specimen was immersed in 0.5% KOH for 48 h, rinsed constantly for 6 h under a weak stream of tap water, and left for 48 h at room temperature in a solution of 35 ml of saturated sodium borate and 65 ml of distilled water. Subsequently, the specimen was dehydrated by soaking it in 90% ethanol until inversion for two hours. Cartilage staining was performed with 0.01 g/L alcian blue in absolute ethanol for 24 h at room temperature, then rinsed with distilled water for 25–30 min and soaked in sodium borate solution for 2 h. The specimen was then immersed in a 1% trypsin solution for 24 h until 25–30% of bone structures were visible and stained with a 0.5% KOH solution supplemented with Alizarin Red S (0.20 g/L) for 24 h at room temperature. Finally, the individual was rinsed with distilled water for 2 h and immersed in a solution of 0.5% KOH and glycerol (2:1 ratio) for 10 days; once the solution was light pinkish-reddish color, it was replaced with a new solution of 0.5% KOH and glycerol (1:1 ratio) for 14 days. The specimen was stored in pure glycerol. Osteological terminology is based on Trueb (1973), Duellman & Trueb (1994), Fabrezi & Alberch (1996), Guayasamin & Terán-Valdez (2009), Scherz et al. (2017), Suwannapoom et al. (2018) and Reyes-Puig et al. (2021). Cartilage structures were omitted from the osteological description.

DNA extraction, amplification, and sequencing. DNA extraction, PCR amplification, and subsequent Nanopore sequencing were performed at the Nucleic Acid Sequencing Laboratory of the Instituto Nacional de Biodiversidad (INABIO) in Quito, Ecuador. DNA was extracted from muscle tissues under standard protocols using the GeneJET Genomic DNA Purification Kit (K0722). Gene amplification was performed by Polymerase Chain Reaction (PCR) for three contiguous mitochondrial genes 12S rRNA, 16S rRNA, and NADH dehydrogenase subunit 1 (ND1), 10 µl reactions consisted in 4.9 µl 2X Dreamtaq Hotstart Mastermix (Thermo Fisher Scientific), 1 µl DNA template (10 ng), 0.8 µl Forward Primer (10 µM), 0.8 µl Reverse primer (10 µM), and 2.5 µl nuclease-free water. We targeted two overlapping fragments. A 2,000 bp fragment of 12S rRNA and a partial fragment of 16S rRNA using 12sL4E and 16H36E (Heinicke, Duellman & Hedges, 2007) and the 2500 bp complementary fragment of 16S rRNA with NADH dehydrogenase subunit 1 gene (ND1) using 16L19 and t-Met-frog (Wiens et al., 2005; Moen & Wiens, 2009). For the first fragment, PCR conditions were 95° for 5 min as initial denaturation followed by 35 cycles of denaturation at 95° for 30 s, annealing at 50° for 30 s, extension at 72° for 200 s, and a final extension of 72° for 5 min. For the second fragment, PCR conditions were 95° for 5 min as initial denaturation followed by 35 cycles of denaturation at 95° for 30 s, annealing at 57° for 30 s, extension at 72° for 230 s, and a final extension of 72° for 5 min. Sequencing was performed on a minION Mk1C using Flongle Flow Cell R9.4.1 and Rapid Barcoding Kit 96 following manufacturer protocols. Raw reads were High-accuracy (HAC) basecalled and demultiplexed using guppy 6.4.6. Basecalled FASTQ reads were filtered at a Qscore of 9, and fasta consensus was generated using NGSpeciesID (Sahlin, Lim & Prost, 2021).

Phylogenetic analyses

A first concatenated matrix was built with newly generated sequences and GenBank sequences for Noblella mitochondrial genes 12S rRNA, 16SrRNA, and ND1. The matrix was visually inspected in Mesquite 3.81 (Maddison & Maddison, 2018) and aligned using MAFFT 7.526 (Katoh et al., 2002). Due to the lack of 12S and ND1 sequences representation among Noblella, our matrix was trimmed to a partial fragment of 1104 bases of the 16S rRNA mitochondrial gene.

Matrix was completed following the phylogeny proposed by Reyes-Puig et al. (2021), and using GenBank sequences of Noblella, Euparkerella, Barycholos, Ischnocnema, Heyerus, Bryophryne, Microkayla, Qosqophryne, and Psychrophrynella originally published by: Lehr, Fritzsch & Müller (2005), Frost et al. (2006), Hedges, Duellman & Heinicke (2008), Padial, Castroviejo-Fisher & De la Riva (2009), Canedo & Haddad (2012), Von May et al. (2017), Lyra, Haddad & De Azeredo-Espin (2017), De la Riva et al. (2017), Reyes-Puig et al. (2019), Catenazzi et al. (2020), Condori et al. (2020), Reyes-Puig et al. (2020), Reyes-Puig et al. (2021) and Motta et al. (2021). The newly generated sequences are available in GenBank as partial mitogenomes of 4.2 kbp approximately, accessions numbers: PP400756 –PP400760. Substitution models and maximum likelihood tree inference were performed using W-IQ-TREE 2.3.4 (Trifinopoulos et al., 2016) under default settings. Branch support was evaluated using 2,000 ultrafast bootstrapping and SH-aLRT tests with 1,000 replicates (Guindon et al., 2010). Uncorrected p genetic distances were calculated using a 562 bp long fragment of the 16S rRNA using Mesquite (Maddison & Maddison, 2018).

We included Craugastor, Pristimantis, Niceforonia, and Bolitoglossa representatives as outgroups to root our phylogenetic tree and provide a comparative framework. Diverse outgroups helped stabilize phylogenetic inference, test evolutionary hypotheses, and ensure accurate interpretation of evolutionary relationships within Noblella. Phylogenetic tree was visualized and edited using FigTree v1.4.4 (Rambaut, 2018) and Adobe Illustrator 27.9.4.

Extinction risk assessment. The assessment was conducted in accordance with the protocol presented by IUCN (2012) and guidelines provided by the IUCN Standards and Petitions Committee (2022). Extent of occurrence (EOO) and area of occupancy (AOO), using a cell size of 2 km, were calculated utilizing GeoCAT (Bachman et al., 2011).

Results

Phylogenetic relationships and genetic distances (Figs. 2 and 3)

The matrix was 1104 bp long for 82 individuals. IQTree evaluated the 16S gene best-fit model as TIM2+F+G4 according to BIC. Our inferred phylogeny is consistent with previous phylogenies, showing a polyphyletic Noblella, with Northern Clade restricted to species of Ecuador and Northern Peru, and all other species in Southern Clade (Reyes-Puig et al., 2020; Reyes-Puig et al., 2021). The new species described herein is part of the Nothern Clade of Noblella (Reyes-Puig et al., 2019; Reyes-Puig et al., 2020; Reyes-Puig et al., 2021), belonging to a well-supported clade with two specimens identified as Noblella myrmecoides from Ecuador and Peru and a candidate species Noblella sp. from San Martin, Peru reported as Phyllonastes sp. by Lehr, Fritzsch & Müller (2005).

Figure 2 Phylogenetic relationships of Noblella arutam sp. nov.

Phylogenetic relationships among Noblella (light gray boxes). In green, Noblella arutam sp. nov. Numbers in branchs are SH-aLRT support (%) / ultrafast bootstrap support (%). Outgroups are not shown. GenBank accession number is shown to the left of the species name and museum catalog number is shown to the right, if available. Figure by Julio C. Carrión-Olmedo.

Figure 3 Uncorrected p distances for Noblella arutam sp. nov. and its closer congeners.

Figure by Julio C. Carrión-Olmedo.

Within our clade of study, uncorrected p distances range from 6.02% to 10.53% as shown in Fig. 3. The new species is placed in the genus Noblella sensu lato based on morphological characters following the definition proposed by Hedges, Duellman & Heinicke (2008) and the genetic evidence detailed herein. Based on external morphology, osteology and phylogenetic evidence, we confirm the validity of Noblella arutam sp. nov. as an independent evolutionary lineage (De Queiroz, 2007).

Systematic account

Noblella arutam new species

Figures 4, 5, 6 and 7

Figure 4 Noblella arutam sp. nov.

Preserved holotype ZSFQ 1882, adult female, SVL = 20.34 mm. (A) Dorsal view; (B) ventral view; (C) lateral view. Photographs by David Brito-Zapata.

Figure 5 Manus and pes of Noblella arutam. sp. nov.

Preserved paratype of Noblella arutam sp. nov. ZSFQ 1875, adult female. (A) Left palmar surface; (B) left plantar surface. Photographs by David Brito-Zapata.

Figure 6 Variation in preserved Noblella arutam. sp. nov.

Preserved individuals of Noblella arutam sp. nov. showing dorsal and ventral views. Males, (A, F) ZSFQ 1885, SVL = 13.84 mm, paratype; (B, G) ZSFQ 6236, SVL = 14.29 mm, paratype; (C, H) ZSFQ 6240, SVL = 14.32 mm, paratype; (D, I) ZSFQ 6241, SVL = 15.1 mm, paratype; (E, J) ZSFQ 6235, SVL = 14.52 mm, paratype. Females, (K, P) ZSFQ 1875, SVL = 17.25 mm, paratype; (L, Q) ZSFQ 6232, SVL = 17.4 mm, paratype; (M, R) ZSFQ 6231, SVL = 20.03 mm, paratype; (N, S) ZSFQ 6233, SVL = 18.52 mm, paratype; (O, T) ZSFQ 6234, SVL = 17.95 mm, paratype. Photographs by David Brito-Zapata.

Figure 7 Variation in life Noblella arutam. sp. nov.

Dorso-lateral, dorsal and ventral patterns of Noblella arutam sp. nov. in life. (A–C) holotype ZSFQ 1882, adult female, SVL = 20.34 mm; (D–F) Paratype ZSFQ 1875, adult female, SVL = 17.25 mm. (G–I) Paratype ZSFQ 6234, adult female, SVL = 17.95 mm. Photographs by Carolina Reyes-Puig and David Brito-Zapata.

LSID urn:lsid:zoobank.org:act:6AF35E9D-3DB0-459B-B7F9-5729F6468899

Proposed standard English name. Arutam Leaflitter Frog

Proposed standard Spanish name. Rana de Hojarasca de Arutam

Holotype. ZSFQ 1882, adult female collected at the Comunidad de Río Blanco (3.9086°S, 78.4892°W, 1850 m), Parroquia de Paquisha, Cantón Paquisha, Provincia de Zamora Chinchipe, República del Ecuador, by David Brito-Zapata and Juan Hurtado on 21 February 2019 (Figs. 1 and 4).

Paratypes (5 ♀, 8 ♂). Adult females collected near the type locality: ZSFQ 1875 (3.9118°S, 78.5028°W, 1720 m), by David Brito-Zapata and Juan Hurtado on 20 February 2019; ZSFQ 6231 (3.9183°S, 78.5047°W, 1750 m), ZSFQ 6233 (3.9177°S, 78.5041°W, 1770 m) by Carolina Reyes-Puig, David Baez, David Brito-Zapata, Elías Figueroa-Coronel, and Juan Hurtado on 10 August 2023; ZSFQ 6232, 6234 (3.9183°S, 78.5047°W, 1750 m) by Juan Hurtado and Andy Hurtado on 12 August 2023. Adult males collected near the type locality: ZSFQ 1189 (3.9094°S, 78.4955°W, 1830 m) by David Brito-Zapata and Juan Hurtado on 12 February 2018; ZSFQ 1885–6 (3.9071°S, 78.4951°W, 1890 m) by David Brito-Zapata and Juan Hurtado on 22 February 2019; ZSFQ 6235–6 (3.9183°S, 78.5047°W, 1750 m), ZSFQ 6241 (3.9177°S, 78.5041° W, 1770 m) by David Brito-Zapata, Carolina Reyes-Puig, Elías Figueroa-Coronel, David Báez, and Juan Hurtado on 10 August 2023; ZSFQ 6238, 6240 (3.9183°S, 78.5047°W, 1750 m) by Juan Hurtado and Andy Hurtado on 12 August 2023.

Etymology. Noblella arutam sp. nov. is named in reference to Arútam, the spiritual entity symbolizing divine power in the world of the Shuar. The Shuar people are native to western Amazonia, spanning Ecuador and Peru. Several Shuar communities reside in the Cordillera del Cóndor and its immediate surroundings, where this new frog species also inhabits. Arútam is believed to dwell in waterfalls, rivers, mountains, rocks, trees, and certain plants and animals (Delgado, 1986; Pellizzaro, 2005). This deep association highlights the profound connection between nature and the Shuar people. Through this name, we pay tribute to the Shuar people living near the type locality of Noblella arutam sp. nov., their ancestral lands and culture, acknowledging their tireless efforts and commitment to nature protection.

Definition. Noblella arutam sp. nov. (Figure 4–7) differs from its congeners by the following characteristics: (1) skin of dorsum shagreen, with scattered low tubercles more evident on flanks; skin on venter smooth, discoidal fold present, thoracic fold slightly defined; (2) tympanic annulus and membrane defined and visible externally, supratympanic fold slightly defined; (3) snout subacuminate in dorsal view, rounded in lateral view; (4) upper eyelids bearing one or two inconspicuous tubercles; (5) vomerine teeth absent; (6) vocal slits and external vocal sac present, nuptial pads absent; (7) fingers not expanded distally, finger tips acuminate more evident on Finger III, papilla absent, Finger I smaller than Finger II; (8) distal phalanges strongly T-shaped, phalangeal formula on hands 2-2-3-3; (9) minute supernumerary palmar tubercles present, slightly visible, ulnar tubercles low and inconspicuous, subarticular tubercles rounded, circumferential grooves absent; (10) heel with one small subconical tubercle; one elongated and subconical tarsal tubercle; two prominent metatarsal tubercles, inner ovoid and outer subacuminated; toes slightly expanded and rounded distally, with papillae in all toes, more evident in toes II–IV; (11) Toe V shorter than Toe III, supernumerary plantar tubercles small and white, circumferential grooves present; (12) phalangeal formula of feet 2-2-3-4-3; (13) in life, dorsum light brown, light brown with gray, or dark brown, with V-shaped inverted or scattered irregular darker marks, with or without inconspicuous middorsal cream stripe from scapular region to cloaca; black suprainguinal marks; black or dark brown lateral band from tip of snout to midbody, with scattered white dots; groins yellow to orange with minute brown dots; venter yellow, dotted with white and brown; (14) SVL in adult males 13.8–15.1 mm (x ¯ 14.6 ± 0.4 mm, n = 8), SVL in adult females 17.3–20.3 mm (x ¯ 18.6 ± 1.3 mm, n = 6).

Diagnosis. Noblella arutam sp. nov. can be easily distinguished from all other species currently assigned to Noblella by the following combination of characters: upper eyelids bearing one or two inconspicuous tubercles, distal phalanges strongly T-shaped, phalangeal formula on hands 2-2-3-3, phalangeal formula on feet 2-2-3-4-3, heel with a small subconical tubercle, disc on all toes with papillae, more evident in toes II–IV and dorsum light brown, light brown with gray or dark brown, with V-shaped inverted or scattered irregular darker marks. Similar to the new species, N. naturetrekii and N. personina (Harvey et al., 2013) occur in the eastern slopes of the Andes of Ecuador. Noblella arutam sp. nov. differs from N. naturetrekii (characters in parentheses) by the presence of three phalanges on Finger IV (two phalanges), skin on venter smooth (granular), nuptial pads absent (present), toe tips with papillae (papillae absent), larger SVL, males13.8–15.1 mm, females 17.3–20.3 mm (males 11.2–12.1 mm, females 13.1–14.4 mm). Noblella arutam sp. nov. is diagnosed from N. personina (characters in parentheses) by having black suprainguinal marks (no suprainguinal marks), dorsal skin shagreen with scattered low tubercles more evident in the flanks (smooth with few low pustules, most dense on posterior half of body), snout subacuminated in dorsal view (rounded). Noblella arutam sp. nov. can be distinguished from the species of Noblella from the western slopes of the Andes of Ecuador and Peru, N. coloma, N. heyeri (Lynch, 1986), N. mindo and N. worleyae, by external morphological characters (characters in parentheses). Noblella arutam sp. nov. has snout subacuminated in dorsal view, N. coloma, N. mindo and N. worleyae (rounded); Noblella arutam sp. nov. has toe tips with papillae, N. mindo, N. heyeri and N. worleyae (without papillae); Noblella arutam sp. nov. differs from N. coloma by having ulnar tubercles (ulnar tubercles absent), in life, dorsum with V-shaped inverted or scattered irregular darker marks (dorsum uniform brown with distinctive suprainguinal marks), venter yellow dotted with white and brown (bright orange); Noblella arutam sp. nov. differs from N. mindo by having fingertips acuminate (rounded), inconspicuous middorsal cream stripe present or not (dorsum with a cream middorsal, longitudinal line distinct and present in all individuals), thoracic fold sligthly defined (absent), throat slightly pigmented by minute brown dots (dark brown); Noblella arutam sp. nov. differs from N. heyeri by having skin of venter smooth (pitted), discoidal fold present (absent), fingertips not expanded (slightly expanded); Noblella arutam sp. nov. differs from N. worleyae by having dorsum with V-shaped inverted or scattered irregular darker marks (dorsum with a middorsal line continuing the posterior lengths of hind legs cream to light brown). We differentiated the new species from N. lochites sensu stricto and N. myrmecoides sensu stricto (characters in parentheses) by having the snout subacuminated in dorsal view (short and truncated in dorsal view), and by having ulnar tubercles (absent); Noblella arutam sp. nov. differs from N. lochites sensu stricto by having discoidal fold (absent); Noblella arutam sp. nov. differs from N. myrmecoides sensu stricto by having three phalanges on Finger IV (two phalanges). Comparisons with other species of Noblella are detailed in Table S1.

Description of holotype. Adult female (ZSFQ 1882) (Fig. 4); head narrower than body, HL 33.9% of SVL; head as long as wide; snout subacuminate in dorsal view and rounded in lateral view; canthus rostralis straight; loreal region slightly concave; MWE 57.3% of MIOD; EN 53.2% of ED; tympanum visible externally, tympanic membrane differenced from surrounding skin, supratympanic fold slightly defined (reduced by preservation effects). Dentigerous processes of vomers absent; choanae laterally oriented; tongue longer than wide, partially notched posteriorly.

Skin of dorsum and flanks shagreen with scattered low tubercles more evident on flanks; venter smooth; discoidal fold slightly visible, dorsolateral folds absent, thoracic fold slightly defined; ulnar tubercles low and inconspicuous; palmar tubercle oval, about 2 × size of thenar tubercle; minute supernumerary palmar tubercles; proximal subarticular tubercles big and rounded; fingers not expanded distally, tip of fingers acuminate more evident on Finger III, circumferential grooves absent; relative lengths of fingers: I <II <IV<III; forearm lacking tubercles.

Hindlimbs length moderated, LS 47% of SVL; foot length 47% of SVL; dorsal surfaces of hindlimbs shagreen, heels with a small subconical tubercle (reduced by preservation effects), one elongate, subconical tarsal tubercle on ventral surface of tarsus; two prominent metatarsal tubercles, inner one ovoid and external subacuminate; toes expanded and rounded distally, toe tips with a papilla more evident and prominent on toes II, III and IV; distal portions of circumferential grooves slightly visible; relative lengths of toes: I <II <V <III <IV.

Measurements of holotype (in mm): SVL = 20.34, HL = 6.9, HW = 6.9, ED = 2.48, EN = 1.32, MWE = 1.56, TD = 1.48, MIOD = 2.72, LH = 4.13, LS = 9.57, LF = 9.56. For measurements of the type series (mm) see Table 1.

Table 1 Measurements (in mm) of type series of Noblella arutam sp. nov. Ranges followed by mean and standard deviation in parentheses.

Characters	Noblella arutam sp. nov.	
	Females (n = 6)	Males (n = 8)	
SVL	17.3–20.3 (18.6 ± 1.3)	13.8–15.1 (14.6 ± 0.4)	
HL	5.8–6.9 (6.5 ± 0.4)	4.3–5.3 (4.9 ± 0.3)	
HW	5.9–6.9 (6.3 ± 0.4)	4.4–5.4 (4.8 ± 0.4)	
ED	2.1–2.5 (2.3 ± 0.2)	1.7–2.0 (1.9 ± 0.1)	
EN	1.3–1.6 (1.4 ± 0.1)	0.9–1.3 (1.1 ± 0.1)	
MWE	1.4–1.6 (1.5 ± 0.06)	1.0–1.4 (1.2 ± 0.1)	
TD	1.1–1.5 (1.3 ± 0.1)	0.8–1.0 (0.9 ± 0.08)	
MIOD	2.2–2.7 (2.5 ± 0.2)	1.7–2.5 (2.1 ± 0.3)	
HaL	3.6–4.1 (3.9 ± 0.2)	2.6–3.1 (2.9 ± 0.2)	
SL	8.6–9.6 (9.0 ± 0.4)	6.1–7.6 (7.0 ± 0.5)	
FL	8.6–9.7 (9.1 ± 0.4)	6.0–7.4 (6.9 ± 0.5)	

Color of holotype in life (Fig. 7). Dorsum brown with darker V-shaped inverted marks and suprainguinal black marks. Dorsal surfaces of forelimbs dark brown with few small cream blotches and black irregular blotches surrounded by light brown; dorsal surfaces of hands brown with light brown and dark brown rounded marks. Dorsal surfaces of hindlimbs brown with darker bars, feet brown with light brown and dark brown rounded marks. Dorsal region of flanks brown, lateral black band with scattered white dots from tip of snout to midbody, getting thinner at posterior end. Groins yellow to orange with minute brown dots; venter and ventral surfaces of limbs yellow dotted with white and brown dots, more concentrated on forelimbs and anterior region of throat; iris black with reddish copper around pupil and scattered small dots of same color.

Color of holotype in ethanol (Fig. 4). Dorsum brown, darker towards the head, suprainguinal black marks. Dorsal surfaces of forelimbs dark brown with scarce small white blotches and black irregular blotches, surrounded by cream, dorsal surfaces of hands light brown with cream and dark brown rounded marks. Dorsal surfaces of hindlimbs brown with darker bars, feet lighter with cream and dark brown rounded marks. Dorsal region of flanks brown; lateral black band from tip of snout to midbody, getting thinner at posterior end. Groins cream with minute brown dots; venter and ventral surfaces of limbs cream with minute brown dots, more concentrated on forelimbs and anterior region of throat.

Variation (Fig. 6). Dorsal coloration varies from light brown (ZSFQ 1875, 6231), light brown with gray (ZSFQ 1885) to dark brown (ZSFQ 6233); with darker V-shaped inverted marks (ZSFQ 1875, 1885, 6231-6235, 6236, 6240), or with scattered darker marks (ZSFQ 6241), specimens ZSFQ 6232, have an inconspicuous middorsal cream stripe from scapular region to cloaca. Black to dark brown band from the tip of the snout to midbody, not evident in ZSFQ 6233. In preservative dorsal coloration is brownish to grayish. Venter and ventral surfaces of limbs yellow with small white blotches, and minute dark brown dots more abundant in the specimen ZSFQ 1885. Specimens ZSFQ 1885, 6233, 6234, 6235, 6241 have dark brown throat due to the accumulation of brown dots. In preservative ventral coloration varies from cream to light brown. Morphometric variation is detailed in Table 1 and raw measurements are detailed in Table S2.

Osteology

Skull (Fig. 8). Skull as long as wide; widest part at meeting point between quadratojugal and maxilla. Rostrum short; distance from anterior edge of frontoparietals to anterior face of premaxilla 13% of skull length. At midorbit, braincase about 30% of skull width. Frontoparietal longer than wider, narrower anteriorly and slightly wider posteriorly; narrowly separated at anterior and wider at posterior region, fused anteriorly with two fontanels, U-shaped crest. Ventrally, prootics in contact with parasphenoid. Prootics well separated. Exoccipitals separated the same distance as ventrally and dorsally. Anterolaterally, frontoparietals in contact with sphenethmoid. Sphenethmoid well-ossified, ventrally fused both in anterior and posterior half. Cultriform process of parasphenoid well ossified posteriorly, anteriorly abroad and 36% width of braincase at mid-orbit. Lateral margins of process approximately parallel. Parasphenoid alae moderated and well-ossified. Neopalatines wide and short, articulated with sphenethmoid but not in contact with maxillae. Collumela long and well-ossified. Poorly developed dorsal investing bones. Nasals thin and quite separate, curved ventrally, very closely approximated posterolaterally towards maxillae. Prevomers small and broadly separated. Maxillary arcade holds several small, poorly developed teeth on premaxillae and maxillae. Premaxillae and its anterodorsal alar processes medially separated and separated from nasals. Premaxilla and maxilla in lateral contact. Maxilla long, posterior end acuminated and in contact with quadratojugal. Quadratojugal very thin, strongly curved and articulating anteriorly with maxilla and posterodorsally with ventral branch of squamosal. T-shaped squamosal with long and thin laminar otic ramus; zygomatic ramus much shorter, almost 1/8 of otic ramus; ventral ramus 2 × length of laminar otic ramus. Mandible slim and edentate. Mentomeckelians small and broadly separated. Dentary long and thin, from about anterior corner of orbit posteriorly acuminate and in contact with angulosplenial bone for most of its length and overlapping, except at most anterior part; anteroventrally in contact with mentomeckelian bones. Angulosplenial long and arched, anteriorly wider than posteriorly. Coronoid process long and squared ridge.

Figure 8 Skull morphology of Noblella arutam. sp. nov.

(A) Dorsal, and (B) ventral views of Noblella arutam sp. nov. ZSFQ 1886, paratype, adult male. col, columella; fpar, frontoparietale; max, maxilla; nas, nasal; npl, neopalatino; occ con, occipital condyle; proot, prootic; exoc, exoccipital; pmax, premaxilla; prsph, parasphenoid; qj, quadratojugal; spheth, sphenethnoid; sq, squamosal. Photographs by David Brito-Zapata.

Postcranium (Fig. 9).

Eight presacral vertebrae, all non-imbricate and exhibiting similar longitudinal characteristics; except for presacral I, all others with well-developed diapophyses. Transverse processes of presacral vertebrae V to VIII uniform in size, presacral III longer, followed by presacral IV; presacral II with the shortest transverse processes, which are relatively wider than other presacral vertebrae. Urostyle lengthy and slender, mirroring length of presacral portion of vertebral column; with a prominent dorsal ridge along its length that decreases towards posterior apex. Pectoral bridge with well-ossified coracoids and clavicles. Scapular partially ossified, and sternum unossified. Pelvic girdle elongated and slender, with dorsal surface of iliac bearing a notable ridge lower on anterior apex.

Figure 9 The full skeleton of Noblella arutam. sp. nov.

(A) Dorsal, and (B) ventral views of Noblella arutam sp. nov. ZSFQ 1886, paratype, adult male. antbr = os antebrachii (radius + ulna), clav = clavicle, cor = coracoid bone, crur = os cruris (tibia + fibula), fem = femorale bone, fib = fibulare, hm = humerale bone, il = ilium, presac v = presacral vertebrae, sac v = sacral vertebrae, sc = scapula, ur = urostyle, tib = tibiale. Photographs by David Brito-Zapata.

Figure 10 Manus and pes osteology of Noblella arutam sp. nov.

The left hand in dorsal view (A), and the right foot dorsal view (B) of Noblella arutam sp. nov. ZSFQ 1886, paratype, adult male. Digits numbered I–V. antbr = os antebrachii (radius + ulna), carp d, carpale distale; cent, centrale; fib, fibulare; mtc, metacarpalia; mtt, metatarsalia; ph d I–IV, finger phalanges F1–F4; ph d I–V, toe phalanges F1–F5; prhl, prehallux; rad, radius; tar d, tarsale distale; tib, tibiale; uln, ulnare. Photographs by David Brito-Zapata.

Manus and Pes (Fig. 10). Phalanges ossified, phalangeal formula for fingers: 2-2-3-3 and for toes: 2-2-3-4-3 . Finger length: I <II <IV <III, and toe length: I <II <V <III <IV. Distal knobs on terminal phalanges of all fingers and toes. Terminal phalanges narrower than penultimate phalanges on all fingers and toes. Carpus with radiale, ulnare, Element Y, and not well-ossified prepollex element (hardly visible in photographs). Tarsus with Tarsal 1 and Tarsal 2 + 3, centrale, and not well-ossified prehallux.

Distribution and natural history. All individuals of Noblella arutam sp. nov. are known from the Río Blanco community and surrounding areas, Paquisha county, Zamora Chinchipe province, Ecuador, at 1720–1890 m elevation, in the Montane Evergreen Forest of the Cordilleras del Cóndor and Kutukú ecosystem (Morales, Jadán & Aguirre, 2013), characterized by its high humidity and abundance of epiphytes, bryophytes, and leaflitter. Most specimens of Noblella arutam sp. nov. were found active at night on the ground, amidst leaflitter, and three individuals were found in the leaflitter but in the morning. No vocalizations of the new species were heard during the expeditions. Four paratypes were collected in pitfall traps. The stomach content of specimen ZSFQ 1886 contained three ant-mimicking rove beetles (Staphylinidae: Pselaphinae), two aquatic fly larvae (Diptera: Nematocera), and eight ants (Formicidae: Myrmicinae). Populations of the new species in the type localities appear locally abundant. Although we included only 14 specimens in the type series, we collected 19 individuals and caught and released many more individuals during the expeditions. The following species were found syntopic: Pristimantis galdi (Jiménez de la Espada, 1870), P. daquilemai (Brito-Zapata et al., 2021), P. prolatus (Lynch & Duellman, 1997), Lynchius simmonsi (Lynch, 1974b), and Rhinella cf. margaritifera (Laurenti, 1768).

Extinction risk. All known records of Noblella arutam sp. nov. are from surrounding areas of the Río Blanco community in the Cordillera del Cóndor, which we considered as a single locality due to their proximity and the fact that they are within an area with the same threat. The main threat to this locality is habitat degradation or loss caused by mining activities because it is within a large-scale mining concession (Roy et al., 2018; Foro Intergubernamental sobre Minería, Minerales, Metales y Desarrollo Sostenible I, 2019). Other anthropogenic activities, such as agriculture and livestock farming, have been identified as threats to the native ecosystems of the Cordillera del Cóndor (Brito-Zapata et al., 2021). However, emigration of people from the Río Blanco community has significantly increased in recent years, leading to reduction in its impacts. We proposed that N. arutam sp. nov. should be classified as Critically Endangered (CR), based on criteria B1ab(iii) +2ab(iii) based on the following assessment of its extinction risk: (1) Extent of occurrence EOO of 1.70 km2, within the threshold for Critically Endangered (<100 km2), criterion B1. (2) Area of occupancy AOO of 8 km2, also within the threshold for Critically Endangered (<10 km2), criterion B2. (3) Known from a single locality. (4) Continuing decline inferred of area of occupancy, extent of occurrence, and quality of habitat due to impacts derived from large-scale mining.

Remarks. Some morphological characters are less noticeable in some specimens due to preservation effects. Tubercles on the dorsum and flanks, supratympanic fold, ulnar tubercles, and heel tubercle are lower and less apparent in some preserved specimens of Noblella arutam sp. nov.

Discussion

Although our study relies only on the widely used 16S ribosomal RNA gene to infer phylogenetic relationships, we opted to utilize the longest fragment possible as full-length 16S rRNA recover the most accurate phylogenetic relationships, highest branch support, lowest variation in genetic distances (pairwise p-distances), and best-scoring species delimitation partitions (Chan et al., 2022). However, we are aware that future integration of additional genetic markers will be essential to our understanding of Noblella evolutionary history and ensuring robust phylogenetic inference (Dietz et al., 2023). Uncorrected p genetic distances between Noblella arutam sp. nov. and other species is bigger than 6% in all cases (Fig. 3), therefore the certainty to validate as new species is higher. In many cases Noblella species have a relatively high percentage of genetic distance, however, this percentage may be less than 6% as in with N. coloma and N. worleyae (Reyes-Puig et al., 2021).

Our phylogeny agrees with previous phylogenetic analyses of Noblella (Catenazzi & Ttito, 2019; Reyes-Puig et al., 2019; Reyes-Puig et al., 2020; Reyes-Puig et al., 2021; Santa-Cruz et al., 2019). It shows two well-supported, not-closely related clades of Noblella. The Southern Clade includes N. losamigos (Santa-Cruz et al., 2019), N. madreselva (Catenazzi, Uscapi & May, 2015), N. pygmea (Lehr & Catenazzi, 2009), N. thiuni (Catenazzi & Ttito, 2019), a potentially undescribed species of Noblella (CORBIDI AC-2019), a specimen identified as N. myrmecoides (RvM3), and species of the genus Psychrophrynella. The Northern Clade is composed by Noblella coloma, N. heyeri, N. lochites, N. mindo, N. naturetrekii, N. personina, N. worleyae, a specimen identified as N. myrmecoides (CORBIDI 9384), two potentially undescribed species of Noblella (ZSFQ 6227 and GF-San Martin) and the new species N. arutam. Specimens identified as N. myrmecoides by Von May et al. (2017), Canedo & Haddad (2012) and Catenazzi & Ttito (2019) are placed in both clades, and considering their genetic and geographical distances we recommend an exhaustive revision of these specimens and the type series of the species to properly identify them.

Species of the genus Noblella are miniaturized and cryptic species (Duellman & Lehr, 2009; Reyes-Puig et al., 2021). We present details of their osteology, based on a stained and cleared specimen of the new species to add evidence of their morphological variation; thus increasing to five species of Noblella with complete osteological description, N. coloma, N. mindo, N. naturetrekii, N. worleyae, and N. arutam sp. nov. (Guayasamin & Terán-Valdez, 2009; Reyes-Puig et al., 2019; Reyes-Puig et al., 2021). Available information suggests that, in addition to the number and shape of phalanges, cranial osteology could be useful to diagnose species.

The area where N . arutam sp. nov. inhabits also shelters several other amphibian species with very restricted distribution: Pristimantis barrigai (Brito & Almendáriz, 2018), P. daquilemai (Brito-Zapata et al., 2021), P. ledzeppelin (Brito-Zapata & Reyes-Puig, 2021), P. paquishae (Brito, Batallas & Velalcázar, 2014). Unfortunately, this area, like many others in the Cordillera del Cóndor, is within large-scale mining concessions, threatening native ecosystems due to deforestation, soil erosion, environmental pollution, and other impacts caused by mining (Roy et al., 2018; Foro Intergubernamental sobre Minería, Minerales, Metales y Desarrollo Sostenible I, 2019). The Cordillera del Cóndor is a highly biodiverse site, not only considering amphibians but also other animals and plants (Brito et al., 2021; Brito-Zapata et al., 2023; Vélez-Abarca et al., 2023). The conservation of native ecosystems of the Cordillera del Cóndor is critical to preserve its impressive biodiversity.

Conclusions

We described a new species of the genus Noblella, based on genetic, morphological, and osteological evidence. N. arutam sp. nov. could be distinguished from its congeners by having two tubercles on the upper eyelids, heel with a small subconical tubercle, disc on toes with papillae, distal phalanges strongly T-shaped, and the Finger IV containing 3 phalanges. Our phylogenetic analysis shows that N. arutam sp. nov. belongs to the northern clade of Noblella. The new species have been registered only in the type locality in the Cordillera del Cóndor, Ecuador and this work contributes to highlight the importance of conserving this area. Its high endemism rate and the continuous description of new species of animals and plants make it a place of high biodiversity.

Supplemental Information

Table S1 Comparison of main characters of Noblella species

Table S2 Morphological measurements of the type series of Noblella arutam sp. nov

We are extremely thankful to Juan Hurtado and his family for their support during fieldwork. We thank David Báez and Elías Figueroa for field assistance, Jorge Montalvo and Margarita López for identifying the insects in the stomach content, and Emilia Peñaherrera for helping with the map and for the administrative work while planning the expedition. We are grateful to the following people for access to collections under their care and support at their respective institutions: Mario Yánez-Muñoz (DHMECN); and George Zug, Roy W. McDiarmid, W. Ron Heyer, Robert Reynolds, Kenneth A. Tighe, Steve W. Gotte, Carole C. Baldwin, and Mary Sangrey (USNM). We thank Diego Inclán, Francisco Prieto, Efraín Freire, and Pablo Jarrín-V. for their institutional support. Daniela Reyes, Pamela Loján, Dayana Vazquez, and Annahy Ayala provided technical assistance during laboratory work.

Additional Information and Declarations

Competing Interests

Author Contributions

Animal Ethics

Field Study Permissions

DNA Deposition

Data Availability

New Species Registration

The authors declare there are no competing interests.

David Brito-Zapata conceived and designed the experiments, performed the experiments, analyzed the data, prepared figures and/or tables, authored or reviewed drafts of the article, and approved the final draft.

Juan D. Chávez-Reyes performed the experiments, authored or reviewed drafts of the article, and approved the final draft.

Matheo David Pallo-Robles performed the experiments, authored or reviewed drafts of the article, and approved the final draft.

Julio C. Carrión-Olmedo performed the experiments, analyzed the data, prepared figures and/or tables, authored or reviewed drafts of the article, and approved the final draft.

Diego F. Cisneros-Heredia performed the experiments, analyzed the data, authored or reviewed drafts of the article, and approved the final draft.

Carolina Reyes-Puig conceived and designed the experiments, performed the experiments, analyzed the data, authored or reviewed drafts of the article, and approved the final draft.

The following information was supplied relating to ethical approvals (i.e., approving body and any reference numbers):

This study was conducted under research permits issued by the Ministerio del Ambiente, Agua y Transición Ecológica, Ecuador (MAAE-ARSFC-2022-2204, 009-2018-IC -FLO-FAU-DPAZCH-UPN-VS/MA, 005-2019-IC-FLO-FAU-DPAZCH-UPN-VS/MA) and framework contract for genetic resources access (MAATE-DBI-CM-2023-0313)

The following information was supplied relating to field study approvals (i.e., approving body and any reference numbers):

This study was conducted under research permits issued by the Ministerio del Ambiente, Agua y Transición Ecológica, Ecuador (MAAE-ARSFC-2022-2204, 009-2018-IC -FLO-FAU-DPAZCH-UPN-VS/MA, 005-2019-IC-FLO-FAU-DPAZCH-UPN-VS/MA) and framework contract for genetic resources access (MAATE-DBI-CM-2023-0313)

The following information was supplied regarding the deposition of DNA sequences:

The newly generated sequences are available in GenBank as partial mitogenomes of 4.2 kbp approximately: PP400756 to PP400760.

The following information was supplied regarding data availability:

The raw data are available in the Supplemental Files.

The following information was supplied regarding the registration of a newly described species:

Publication LSID: urn:lsid:zoobank.org:pub:AD15D34E-81A6-4666-AF30-E53D4F3C5541

Noblella arutam sp. nov. LSID: urn:lsid:zoobank.org:act:6AF35E9D-3DB0-459B-B7F9-5729F6468899

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
