# Peer review of "A new species of frog of the genus Noblella Barbour, 1930 (Amphibia: Strabomantidae) from the Cordillera del Cóndor, Ecuador"

_PeerJ, doi:10.7717/peerj.17939_

## Round 0.1 · original submission · Minor Revisions

Dear Authors,

Thank you for submitting your manuscript titled "A new species of frog of the genus Noblella Barbour, 1930 (Amphibia: Strabomantidae) from the Cordillera del Cóndor, Ecuador" to PeerJ. We have now received and reviewed the comments from our reviewers. Based on their thorough evaluations, we believe that your manuscript has merit and is suitable for publication following minor revisions.

We appreciate your cooperation and look forward to receiving the revised version of your manuscript. Please address the reviewers' comments and provide a detailed response to each point. Once revised, your manuscript will undergo a final review to ensure all issues have been adequately addressed.

Thank you for considering PeerJ for the publication of your work.

Best regards,

Armando Sunny

Reviewer 1 ·

Basic reporting

The authors describe a new frog species in the genus Noblella. The article is well written and provides morphological and molecular data to support the recognition of the new taxon. However, the authors need to provide more information before the manuscript is ready for publication.
Table 1 – what do the abbreviations mean?
- Please add a figure description explaining what the coloured and grey geographic areas in the map are, or add country/region names.
- Figures are not mentioned in the order they are numbered, Fig 2 is mention before Fig 1.; Fig 9 is mentioned before Fig 8. Please reorganize accordingly.
- Figure 1. Add accession numbers to N. arutam and ZSFQ6227 Noblella sp. branch tips. Why do you use “Noblella” for the northern clade and Noblella (without quotes) for the southern clade?

Experimental design

- Could you do a comparative analysis with the morphological character information you gathered for N. arutam and other species Noblella spp? A clustering analysis would support better your conclusion recognizing this taxon as a new species.
- Please at least discuss the implications of using only one marker in you phylogenies, especially a mtDNA marker. I applaud your effort for sequencing several mtDNA regions, however nuclear markers are needed (and will be needed) to solve the phylogenetic issues of the species in this genus.
- For the extinction risk assessment you considered a reduced distribution range. How well explored are the surrounding areas to be sure that N. arutam does not live anywhere else. Perhaps you could include in your map the distribution range and the explored areas. How are the ranges of other species in the genus?
- Please add initial or final concentration of PCR reagents in the reaction (e.g primers) and PCR temperature and cycle conditions.
- add software versions (e.g. IQTree, Mesquite, MAFFT, etc.). How did you visualize and edited the tree?

Validity of the findings

- You mentioned that you assume the unified species concept proposed by de Queiroz (2007), however, you do not discuss how your description fall within this concept.

Additional comments

- I consider that the terms ‘Northern’ and ‘Southern’ Noblella clades are misleading as they are not really sister clades. I understand, after checking the phylogenetic tree and previous research that the genus is not monophyletic, but the authors don’t mention this in the discussion, only refer to previous work. Please, for the reader recently interested in Noblella add some information regarding the phylogenetic uncertainty of the genus.

Reviewer 2 ·

Basic reporting

No comment

Experimental design

The research is within Aims and Scope of the journal. The objective of the Research is well defined, also relevant and meaningful, throughout the description of a new taxon. The presentation of a new species represents an effort to document the biodiversity of an important group of animals. The investigation performed an appropriated technical and ethical standard. Methods are described in general with detail and the necessary information to replicate the study is placed. However, there are a few questions:
Line 182: Why three genes were sequenced for the new species if there is information for species in the genus for 12s only?
Line 185: please clarify that sequences trimmed correspond to partial sequence of the 16s.
Line 195: IQTREE is a fast-searching software for phylogenomic analyses, why is used to analyze a little database of 1104 bp and 82 individuals?
Line 197: The uncorrected P genetic distance was calculated on the base of 562 bp, why these pb were selected?
Line 216: Why are presented only some Uncorrected p distances comparing the new species and three taxa associated in the same clade? but there is another clade associated (Noblella sp and Noblella heyeri), could you present the values for these other two taxa please?
Line 429: 19 individuals were caught, but many more were released, so what was the criterion for maintain the 19 specimens, size, sex, coloration?
Line 457: There is not discussion about pairwise distance results, values were used as barcoding or something like that? there are other groups of species that support your findings?

Validity of the findings

No comments

Reviewer 3 ·

Basic reporting

Brito-Zapata et. al., describe a new species of frogs from the Cordillera del Condor. The manuscript is well-written. The evidence present in the manuscript supports the description of new species.

Experimental design

The study is well-designed and provides a variety of evidence, with good figures, analyses, and discussion of the validity of this new species. However, I am curious about the genetic distances among closely related species, it might be added as supplementary material.

Validity of the findings

I consider this work important to reveal the hidden species diversity within the genus.

Additional comments

Minor comments:

Line 41. Cut off the quotes
Lines 199. Describe briefly why you use four different genera of amphibians as outgroups.
Figure 1. To include the molecular marker used and the number of pair bases.
Figure 2. Adding provinces' names or countries might be helpful.
Figures 7, 8, 9. I suggest creating a single panel that comprises these three figures.

---

## Round 0.2 · accepted · Accept

Dear Authors,

After reviewing the observations from our reviewers, I am pleased to inform you that your manuscript has been accepted for publication. However, there is a minor correction suggested by one of the reviewers:

In Figure 3, please verify the percentage of p distances. It should be 0% instead of 0.27%. This is a recommendation from a reviewer, but I leave it to your discretion.

Thank you very much for choosing PeerJ for the publication of such an interesting work.

Best regards,

Armando Sunny

Reviewer 1 ·

Basic reporting

The manuscript is well written and the changes are well framed and coherent.

Experimental design

No comment

Validity of the findings

No comment

Additional comments

The authors have responded to the issues raised in the previous version.

Reviewer 3 ·

Basic reporting

I think that the authors have adequately addressed the concerns raised during the initial review.

Experimental design

Done

Validity of the findings

The authors have adequately addressed the concerns raised during the initial review. Just one minor comment:
Figure 3. Please check out the percent of p distances, it should be 0% instead 0.27%.